# Cytotoxic Mechanism of Momilactones A and B against Acute Promyelocytic Leukemia and Multiple Myeloma Cell Lines

**DOI:** 10.3390/cancers14194848

**Published:** 2022-10-04

**Authors:** La Hoang Anh, Vu Quang Lam, Akiyoshi Takami, Tran Dang Khanh, Nguyen Van Quan, Tran Dang Xuan

**Affiliations:** 1Transdisciplinary Science and Engineering Program, Graduate School of Advanced Science and Engineering, Hiroshima University, Hiroshima 739-8529, Japan; 2Department of Internal Medicine, Division of Hematology, Aichi Medical University School of Medicine, Nagakute 480-1195, Japan; 3Agricultural Genetics Institute, Pham Van Dong Street, Hanoi 122000, Vietnam; 4Center for Agricultural Innovation, Vietnam National University of Agriculture, Hanoi 131000, Vietnam; 5Center for the Planetary Health and Innovation Science (PHIS), The IDEC Institute, Hiroshima University, Higashi-Hiroshima 739-8529, Japan

**Keywords:** momilactones, leukemia, multiple myeloma, cytotoxic mechanism, apoptosis, cell cycle

## Abstract

**Simple Summary:**

Though the anticancer potentiality of momilactones has been reported in several studies, their cytotoxic mechanism has not been comprehensively scrutinized. In this study, we investigated the cytotoxicity of momilactones A (MA) and B (MB) against acute promyelocytic leukemia (APL) HL-60 and multiple myeloma (MM) U266 cell lines. According to MTT results, MB and the mixture MAB (1:1, *w*/*w*) show a substantial inhibition on the cell viability of HL-60 and U266, with IC_50_ ranging from 4.49 to 5.59 µM. Besides, MB and MAB at 5 µM inhibit HL-60 cells through the regulations of relevant proteins to apoptosis-inducing factors (p-38, BCL-2, and caspase-3) and cell cycle arrest at G2 phase (p-38, CDK1, and cyclin B1). Meanwhile, these compounds enhance U266 apoptosis by altering p-38, BCL-2, and caspase-3 signaling pathways. Significantly, momilactones exhibit a minor effect on a non-cancerous cell line (MeT-5A), implying that they are promising candidates for developing novel anti-APL and anti-MM medicines.

**Abstract:**

This is the first study clarifying the cytotoxic mechanism of momilactones A (MA) and B (MB) on acute promyelocytic leukemia (APL) HL-60 and multiple myeloma (MM) U266 cell lines. Via the MTT test, MB and the mixture MAB (1:1, *w*/*w*) exhibit a potent cytotoxicity on HL-60 (IC_50_ = 4.49 and 4.61 µM, respectively), which are close to the well-known drugs doxorubicin, all-trans retinoic acid (ATRA), and the mixture of ATRA and arsenic trioxide (ATRA/ATO) (1:1, *w*/*w*) (IC_50_ = 5.22, 3.99, and 3.67 µM, respectively). Meanwhile MB, MAB, and the standard suppressor doxorubicin substantially inhibit U266 (IC_50_ = 5.09, 5.59, and 0.24 µM, respectively). Notably, MB and MAB at 5 µM may promote HL-60 and U266 cell apoptosis by activating the phosphorylation of p-38 in the mitogen-activated protein kinase (MAPK) pathway and regulating the relevant proteins (BCL-2 and caspase-3) in the mitochondrial pathway. Besides, these compounds may induce G2 phase arrest in the HL-60 cell cycle through the activation of p-38 and disruption of CDK1 and cyclin B1 complex. Exceptionally, momilactones negligibly affect the non-cancerous cell line MeT-5A. This finding provides novel insights into the anticancer property of momilactones, which can be a premise for future studies and developments of momilactone-based anticancer medicines.

## 1. Introduction

Blood cancer is a serious human disorder, accounting for over 1.2 million cases annually in the world [1]. Among blood cancer types, the incidence of worldwide leukemia was reported as 474,519 new cases with 311,594 deaths in 2020 [1]. Acute promyelocytic leukemia (APL) regularly becomes aggravated during chemotherapy and has a poor prognosis with a high level of early death because of bleeding from coagulopathy. On the other hand, multiple myeloma (MM) occurred in 176,404 cases with 117,007 deaths in 2020 [1]. In MM patients, the accumulated cells in bone marrow can lead to bone lesions with the disruption of structure and function [2]. APL and MM have been becoming serious and complicated problems over the years without any signal of stopping. Therefore, pharmaceutical and medicinal candidates are urgently needed to develop effective treatments for patients suffering from these cancers.

In anticancer studies, numerous strategies have been conducted to promote the apoptotic process, which is a natural mechanism for cell death to control or eliminate the undisciplined expansion of tumors [3]. Enhanced apoptosis is one of the most effective approaches for developing specific anticancer therapy and represents the most successful non-surgical treatment for all cancer cases [3]. Another potential target in anticancer research is the cell cycle, which strictly regulates cell division through multiple control mechanisms [4]. Cancer-associated mutations lead to abnormal regulation that prevents cells from exiting the cell cycle, followed by continuous cell division [4]. Therefore, inducing cell cycle arrest is a promising method for inhibiting tumor proliferation and expansion. Interestingly, both apoptotic and cell cycle processes can be regulated by regulatory proteins [3,5]. Therefore, substances with synergistic effects on apoptosis induction and cell cycle arrest through mediating the activities of relevant proteins may be excellent candidates for developing efficient cancer therapies.

In recent years, a vast number of studies have been conducted considering the anticancer potentials of plant-based products [6,7,8,9], which have exhibited benefits for therapeutic purposes with less toxicity than synthetic medicines [10]. Reality also shows that the simultaneous use of herbal remedies and modern medicine has brought certain effectiveness to the treatment [11]. Among valuable plant-derived analytes, momilactones, diterpene lactones, have been found only in rice (*Oryza sativa*) and the *Hypnum* moss (*Hypnum plumaeforme*). These compounds were first known as phytoalexins, which principally play a role in the defense system of rice against pathogens [12]. Recently, momilactones have exhibited antioxidant, anticancer (leukemia [13], lymphoma [14], and colon cancer [15]), anti-diabetes [16,17], anti-obesity [17], and anti-skin aging properties [18]. Hitherto, the mechanism of cytotoxic and anticancer actions of momilactones has not been comprehensively scrutinized. The limitation of in-depth studies about the anticancer activity of momilactones may be due to the confined availability on the market as well as the difficulty in isolation and purification [16]. Our laboratory is one of the few laboratories in the world that can purify momilactones from natural sources. In preceding reports, we successfully established a method to achieve a remarkable amount of momilactones A and B from rice by-products [16]. 

The aforementioned rationales prompted us to investigate the cytotoxic mechanisms of momilactones A (MA) and B (MB) and their mixture MAB (1:1, *w*/*w*) on HL-60 (a typical cell line isolated from APL patients) and U266 (a well-known cell line derived from MM patients) through apoptotic and cell cycle pathways, and the expressions of relevant regulatory proteins.

## 2. Materials and Methods

### 2.1. Materials

Momilactones A (MA) and B (MB) were previously isolated and purified from rice husk in our laboratory of Plant Physiology and Biochemistry, Hiroshima University, Japan [16]. Briefly, MA and MB were isolated from the ethyl acetate (EtOAc) extract of rice husks (*Oryza sativa* var. Koshihikari) by column chromatography over silica gel with the mobile phase of hexane:EtOAc (8:2). The identification and confirmation of such pure compounds applying TLC, HPLC, LC-ESI-MS, GC-MS, ^1^H-NMR, and ^13^C-NMR were described in the previous study of Quan et al. [16].

The cell lines, including non-cancerous MeT-5A (CRL-9444™), acute promyelocytic leukemia HL-60 (CCL-240™), and multiple myeloma U266 (number: TIB-196™), were purchased from ATCC (Manassas, VA, USA).

### 2.2. Cell Viability (MTT) Assay

In this assay, culture media was prepared by adding fetal bovine serum (10%), L-glutamine (5 mM), penicillin (100 IU/mL), and streptomycin (100 µg/mL) to IMDM (Sigma-Aldrich, St. Louis, MO, USA). The cells (5 × 10^3^ cells/well) were seeded into a 96-well plate filled with 100 µL of culture media and placed in a CO_2_ incubator at 37 °C. After 24 h, the cells were treated with MA, MB, and MAB with different concentrations (0.5, 1, 5, and 10 µM) for 48 h. Subsequently, 10 µL of the MTT solution (5 mg/mL, Sigma-Aldrich) was pipetted into each well. The cells were continuously incubated for 4 h. Finally, 100 µL of cell lysis buffer (10% SDS in 0.01 M HCl) was applied to dissolve the colored formazan crystals. Culture media instead of momilactones was used as the negative control. Meanwhile the drugs consisting of doxorubicin, all-trans retinoic acid (ATRA), the mixture of ATRA and arsenic trioxide (ATRA/ATO) (1:1, *w*/*w*), and bortezomib were tested as the standard inhibitors. The absorbance at 595 nm was scanned to determine the cell growth rate using a spectrophotometer (SpectraMAX M5, Molecular Devices, Sunnyvale, CA, USA) [8]. All tests were performed with three replications. The cytotoxic activity (% inhibition) of momilactones and/or inhibitors on the tested cell lines was as follows:Inhibition (%) = (A_NC_ − A_S_)/A_NC_ × 100(1)
where A_NC_: absorbance of reaction with negative control, and A_S_: absorbance of reaction with momilactone and/or inhibitor.

Dose-responding curves and IC_50_ values (the required concentration for inhibiting 50% of cell viability) of momilactones and the standard inhibitors for cytotoxicity against tested cell lines were established. A lower IC_50_ indicates a stronger cytotoxic activity.

### 2.3. Cell Apoptosis (Annexin V) Assay

The procedure was conducted in triplicate following Lam et al. [9]. In brief, the cells (5 × 10^5^ cells/well) were seeded into a 6-well plate filled with 1.5 mL of culture media and cultured in a CO_2_ incubator for 24 h with the same condition as mentioned in the MTT assay. The cells were then treated with momilactones at a concentration of 5 µM for 24 and 48 h. The non-treated cells were used as a control. Harvested cells were washed twice with cold phosphate-buffered saline (PBS). After that, the control and treated cells were incubated with annexin V-conjugated fluorescein isothiocyanate (FITC) (Biolegend, San Diego, CA, USA) and propidium iodide (PI) for 15 min at 25 °C. The obtained cells were dissolved in 450 µL of PBS. The solution was filtered by a nylon membrane to prevent cell clumping and kept on ice until analysis. The intensities of annexin V-FITC and PI and the percentages of apoptotic cells were instantly determined by a flow cytometer (BD, Franklin Lakes, NJ, USA).

### 2.4. Cell Cycle Assay

The cells (5 × 10^5^ cells/well) were cultured and treated in triplicate following the same methods as the apoptosis assay. The FxCycle PI/RNase staining solution was applied according to the manufacturer’s instructions (Calbiochem, Darmstadt, Germany). The cell cycle distribution at each phase of G1, S, and G2 was determined based on the cell’s DNA content. The percentages of cells in different phases of the cell cycle were quantified by a flow cytometer (BD, Franklin Lakes, NJ, USA). In brief, the collected cells were washed with ice-cold PBS. Subsequently, ice-cold PBS in pure ethanol was added to disperse the cells. The obtained solution was stored at 4 °C for 24 h for fixing. For analysis, the cells were incubated with 10 mg/mL of RNase A (Sigma-Aldrich) for 5 min on ice. The following step was conducted by adding 1 mg/mL of PI (in PBS). After incubating for 10 min at room temperature, the cells were dissolved in 450 µL of PBS. The solution was then filtered using a nylon membrane to remove cell clumping before analysis. Subsequently, the flow cytometric measurement was immediately performed.

### 2.5. Western Blotting Assay

The cells (5 × 10^5^ cells/well) were cultured and treated in triplicate following the same methods as the apoptosis and cell cycle assays. The cell lysates were conducted by rinsing cells with PBS, followed by adding 2× loading buffer two times (4% SDS, 10% 2-mercaptoethanol, 20% glycerol, 0.004% bromophenol blue, 0.125 M Tris-HCl, pH 6.8). The extracted protein (200 pg) was subjected to sodium dodecyl sulfate (SDS)-polyacrylamide gels, applying 10% acrylamide, and subsequently transferred to a polyvinylidene fluoride membrane (Takara Bio, Shiga, Japan) by electroblotting. The membrane was blocked using 3% skim milk in PBS-0.05% Tween 20 (PBS-T) at 25 °C for 1 h. The incubation with each antibody (2 µg/mL) against anti-rabbit total p-38/MAPK, phosphorylated p-38/MAPK, BCL-2, procaspase-3, cleaved caspase-3, CDK1/cdc2, cyclin B1, and GAPDH (BioLegend, San Diego, CA, USA) in blocking buffer was conducted overnight at 4 °C. The following step was performed by washing the membrane with PBS-T in triplicate. The collected membrane was incubated with a secondary antibody of horseradish peroxidase-labeled goat anti-rabbit IgG (20 ng/mL) (IBL, Gunma, Japan) at 37 °C for 1 h. Protein bands were visualized with the use of the LAS-4000 image analyzer (GE Healthcare, Tokyo, Japan). The relative expression (RE) was calculated by normalizing the intensity of targeted proteins to the intensity of the housekeeping protein GAPDH.

### 2.6. Statistical Analysis

Data are displayed as mean ± standard deviation (SD) (*n* = 3). Student’s *t*-test and one-way ANOVA were used to compare differences between groups. The statistical significances were considered at values of *p* < 0.05 (Minitab 16.0 software, Minitab Inc., State College, PA, USA).

## 3. Results

### 3.1. Effects of Momilactones on Cell Viability of Non-Cancerous (MeT-5A), Acute Promyelocytic Leukemia (HL-60), and Multiple Myeloma (U266) Cell Lines

The cytotoxic activities of momilactones A (MA) and B (MB) and their mixture (MAB) (1:1, *w*/*w*) in increased concentrations against the cell viability of non-cancerous MeT-5A, acute promyelocytic leukemia (APL) HL-60, and multiple myeloma (MM) U266 cell lines after 48 h of treatments are displayed in Figure 1. In addition, the cytotoxicity of momilactones is compared with that of well-known medicines, including doxorubicin, all-trans retinoic acid (ATRA), the mixture of ATRA and arsenic trioxide (ATRA/ATO) (1:1, *w*/*w*), and bortezomib.

According to Figure 1 and Table 1, MA, MB, and MAB exhibited a minor inhibition on normal cell line MeT-5A with percentages of 28.52%, 38.00%, and 37.82%, respectively, which are lower than doxorubicin (inhibition percentage = 49.23%) at a concentration of 10 µM (Figure 1). 

The APL cell line HL-60 was least inhibited by MA, with a percentage of 30.25% at 10 µM (Figure 1). Meanwhile, MB and MAB displayed a potent cytotoxic capacity against HL-60 (IC_50_ = 4.49 and 4.61 µM, respectively) which was more substantial than doxorubicin (IC_50_ = 5.22 µM) (Table 1 and Figure 1). The IC_50_ values of ATRA and ATRA/ATO against HL-60 were 3.99 and 3.67 µM, respectively (Table 1 and Figure 1). 

In the case of the MM cell line, MA exhibited the lowest effect on U266 cell proliferation (inhibition percentage = 40.97%) at 10 µM (Figure 1). On the other hand, bortezomib revealed an outstanding prevention against U266 (IC_50_ = 0.008 µM) (Appendix A), followed by doxorubicin, MB, and MAB (IC_50_ = 0.24, 5.09, and 5.59 µM, respectively) (Table 1 and Figure 1).

In general, MA was the weakest compound inhibiting tested cancer cell lines, while MB and MAB substantially suppressed these cell lines at around 5 µM. Therefore, MB and MAB at a concentration of 5 µM were selected for further investigation of their cytotoxic mechanism.

### 3.2. Apoptosis-Inducing Activities of Momilactones against Non-Cancerous (MeT-5A), Acute Promyelocytic Leukemia (HL-60), and Multiple Myeloma (U266) Cell Lines

In this assay, the annexin V method was applied to evaluate the effects of MB and MAB at 5 µM on the cell apoptosis of MeT-5A, HL-60, and U266 cell lines (Figure 2).

The effects of MB and MAB at 5 µM on cell apoptosis of normal cells (MeT-5A) are presented in Figure 2a. The results obtained that after 24 h, MB and MAB revealed a mild decrease in cell apoptosis of MeT-5A (% apoptosis = 2.87% and 2.44%, respectively), while apoptotic cells in the non-treated control accounted for 3.86% (Figure 2a). After 48 h, a slight increase in cell apoptosis of MeT-5A was recorded under the influences of MB and MAB (2.67- and 1.23-fold, respectively) compared to the non-treated control (Figure 2a).

Regarding the APL cell line, MB and MAB revealed an insignificant increase in the apoptotic process of HL-60 after 24 h (1.30- and 3.26-fold, respectively, over the control) (Figure 2b). Remarkably, after 48 h, the number of HL-60 apoptotic cells was dramatically enhanced to 40.50% and 42.10% under MB and MAB effects, respectively, which were much higher than the control (% apoptosis = 0.86%) (Figure 2b).

For the tested MM cell line, MB and MAB remarkably promoted U266 cell apoptosis after 24 and 48 h. Apoptotic cells accounted for 4.06%, 16.90%, 22.70%, 18.0%, and 20.5% in the control and the treatments of MB-24 h, MAB-24 h, MB-48 h, and MAB-48 h, respectively (Figure 2c).

In general, MB and MAB promoted apoptosis in cancer cells (HL-60 and U266), but they exhibited just a minor effect on normal cells (MeT-5A).

### 3.3. Effects of Momilactones on Inducing Cell Cycle Arrest of Non-Cancerous (MeT-5A), Acute Promyelocytic Leukemia (HL-60), and Multiple Myeloma (U266) Cell Lines

The effects of MB and MAB at 5 µM on the cell cycle of MeT-5A, HL-60, and U266 cell lines after 24 and 48 h are displayed in Figure 3.

In Figure 3a, MB and MAB revealed a negligible impact on sub-G1 phase of normal cells (MeT-5A) after 24 h. After 48 h, the cell percentages of sub-G1 were increased by 2.59 and 2.65 times under the effects of MB and MAB, respectively, after 48 h, compared to the untreated control. The outcomes revealed that MeT-5A cell death was slightly elevated in the treatment with MB and MAB for 48 h, which is consistent with the apoptosis (annexin V) results, whereas the cell counts had insignificant changes in G1 and G2 phases after 24 and 48 h. Decreased cell numbers were recorded in S phase of the cells affected by MB and MAB. This finding implies that MB and MAB have a trivial effect on the cell cycle of MeT-5A (Figure 3a).

For HL-60, MB- and MAB-treated cells were significantly accumulated in G2 phase of the cell cycle. Particularly, the cell counts increased from 13.9% in G2 phase of the control cells to 30.1% and 52.8%, respectively, in the cells affected by MB and MAB after 24 h (Figure 3b). After 48 h, the percentages of HL-60 cells in G2 phase were 29.2% and 27.7% under the effects of MB and MAB, respectively (Figure 3b). The results indicate that MB and MAB remarkably arrested the cell cycle of HL-60 at G2 phase, accompanied by the reduced percentages of cells in G1 and S phases.

In the case of U266 cells, there was no remarkable difference in the cell number of G1 phase among the control and the treatments with MB and MAB (Figure 3c). Meanwhile, the cell counts were reduced in S and G2 phases, which may be caused by the increased cell deaths in sub-G1 phase under the effects of MB and MAB after 24 and 48 h (Figure 3c). Generally, MB and MAB have negligible impacts on the U266 cell cycle (Figure 3c).

Overall, MB and MAB show a negligible effect on the cell cycle of normal cells (MeT-5A) and MM cells (U266), whereas these compounds activate G2 arrest in the cell cycle of APL cells (HL-60).

### 3.4. Effects of Momilactones on Expressions of Proteins Related to Apoptosis Induction and Cell Cycle Arrest of Acute Promyelocytic Leukemia (HL-60) and Multiple Myeloma (U266) Cell Lines

Based on the results of apoptosis induction and G2 phase arrest in the cell cycle of HL-60 treated with MB and MAB, the expressions of relevant proteins to apoptosis (total p-38, phosphorylated p-38, BCL-2, procaspase-3, and cleaved caspase-3) and G2 phase (total p-38, phosphorylated p-38, CDK1, and cyclin B1) were evaluated (Figure 4a). Meanwhile, MB and MAB promoted the apoptotic process in U266, but failed to activate cell cycle arrest. Therefore, the expressions of regulatory proteins in the apoptotic pathways comprising total p-38, phosphorylated p-38, BCL-2, procaspase-3, and cleaved caspase-3 were examined (Figure 4b).

According to Figure 4a, the expression of phosphorylated p-38/total p-38 was increased in APL (HL-60) cells affected by MB and MAB, and the relative expression (RE) values in the control and the treatments including MB-24 h, MB-48 h, MAB-24 h, and MAB-48 h were 0.48, 1.15, 0.96, 0.82, and 0.85, respectively. Meanwhile, the protein bands of BCL-2 were dramatically degraded under the effects of MB (the RE values in 24 and 48 h treatments were 1.92- and 2.32-fold, respectively, lower than the control) and MAB (the RE values in 24 and 48 h treatments were 2.16- and 2.67-fold, respectively, lower than the control) (Figure 4a). Besides, the RE values of cleaved caspase-3/procaspase-3 were substantially enhanced in HL-60 treated with MB (RE = 0.56 and 0.77 after 24 and 48 h, respectively) and MAB (RE = 2.97 and 5.17 after 24 and 48 h, respectively), while the value in the untreated control was 0.17 (Figure 4a). In the cell cycle pathway, cyclin B1 and CDK1 expressions were remarkably impeded in HL-60 affected by MB and MAB. The RE values in the control and the treatments comprising MB-24 h, MB-48 h, MAB-24 h, and MAB-48 h were 1.17, 0.80, 0.69, 1.39, 0.79, respectively, for cyclin B1, while the values were 1.55, 1.10, 0.75, 0.68, 0.40, respectively, for CDK1 (Figure 4a).

For the MM cell line U266, MB and MAB remarkably elevated the expression of phosphorylated p-38/total p-38 (RE = 1.00, 2.47, 2.35, 1.47, and 1.88 in the control and the treatments of MB-24 h, MB-48 h, MAB-24 h, and MAB-48 h, respectively) (Figure 4b). Besides, MB and MAB inhibited BCL-2 expression after 24 h by 2.37 and 2.74 times, respectively, compared to the control. Meanwhile, after 48 h, MB and MAB decreased the expression of BCL-2 by 4.41- and 2.88-fold, respectively, over the control (Figure 4b). The expression of cleaved caspase-3/procaspase-3 in U266 treated with MB after 24 and 48 h was increased by 5.33 and 8.83 times, respectively, over the control. While in the treatment with MAB, the RE values of caspase-3/procaspase-3 after 24 and 48 h were 6.92 and 9.08 times, respectively, higher than the control (Figure 4b).

## 4. Discussion

In anticancer research, the cell viability (MTT) assay is an indispensable initial step in exploring antitumor candidates [6,7,8,9]. Principally, the cytotoxicity of natural compounds is dose-dependent [6,7,8,9]. Therefore, we examined the activities of momilactones in increased concentrations against APL (HL-60) and MM (U266) cell lines, compared with non-cancerous (MeT-5A) cell lines. The drugs, including bortezomib, all-trans retinoic acid (ATRA), arsenic trioxide (ATO), and doxorubicin, were tested as standard suppressors (Table 1, Figure 1, and Appendix A), among which bortezomib is a common medicine for MM patients [19]. ATRA and the ATRA/ATO combination are commonly applied to treat APL [20]. Doxorubicin shows effectiveness in lymphoma and MM therapies [21]. Besides, doxorubicin was widely used as a positive control in the research on cytotoxic activity against the HL-60 cell line [22]. In searching for ideal candidates for the development of novel anticancer medicines, the potential compounds should be able to eliminate cancer cells with an IC_50_ of less than or equal to 5 µM [23,24]. Accordingly, MB and MAB with an IC_50_ of around 5 µM may be promising substances for developing anti-APL and anti-MM drugs. Remarkably, their cytotoxic abilities were stronger than that of doxorubicin and in line with that of ATRA in preventing HL-60 cells via the MTT assay (Figure 1 and Table 1). Moreover, the candidate must have a high cytotoxic selectivity against tumors without damaging normal cells [25]. Interestingly, these compounds were less toxic to the non-cancerous cell MeT-5A than doxorubicin (Figure 1 and Table 1). Based on that, we selected MB and MAB to clarify their cytotoxic mechanism in suppressing HL-60 and U266 cancer cells.

Via apoptosis, cell cycle, and Western blotting analyses (Figure 2, Figure 3 and Figure 4), MB and MAB may inhibit tested cancer cells through apoptosis induction and cell cycle arrest by regulating relevant protein expressions. Among targeted proteins, the p-38 group serves as an important signaling mediator in the mitogen-activated protein kinase (MAPK) pathway, which contributes to many biological processes, including inflammation, cell cycle, apoptosis, development, differentiation, senescence, and tumor formation in specific cells [26]. Remarkably, previous studies focusing on the regulation of p-38 were conducted to overcome the drug resistance and improve the suppressive effects on MM cell lines consisting of MM.1S, RPMI8226, and U266 [27]. The activation of p-38 can be determined by the fold increase in the expression of phosphorylated p-38/total p-38 [22]. Accordingly, the upregulated phosphorylation of p-38 observed from Western blotting outcomes may cause apoptosis enhancement of HL-60 and U266 treated with MB and MAB (Figure 4). On the other hand, the elevated expression of p-38 can destabilize cdc25b and cdc25c, which subsequently disrupt CDK1/cyclin B1 complex [28]. This disruption may lead to HL-60 cell cycle arrest at G2 phase. Regarding apoptosis, an anti-apoptotic member, namely BCL-2, suppresses the apoptotic process by sequestering the preforms of fatal cysteine proteases or blocking the release of mitochondrial cell death factors into the cytoplasm [29]. Significantly, elevated expression of BCL-2 has been detected in more than half of all cancer cases [3]. Thus, apoptosis promotion by inhibiting BCL-2 activities could be a promising approach to eliminating tumors. Interestingly, numerous plant-based products have shown their role in activating cancer cell apoptosis through the BCL-2 pathway, for example, curcumin from *Curcuma longa* or graviola from *Annona muricata* [3], suggesting a great potential in the development of anticancer medicines. In this study, MB and MAB extremely impeded the expression of BCL-2 in HL-60 and U266 after 48 h (Figure 4), which may motivate the apoptotic process in these cells. Especially, the U266 cell line harboring *t*(11;14) translocation belongs to a MM cytogenetic subgroup, which presents a high level of BCL-2 relative to MCL-1 or BCL-X_L_ [30]. Therefore, MB and MAB with BCL-2 inhibitory effects may be potential candidates for prospective anticancer research on *t*(11;14) MM. As another concern, many traditional medicines suppress cancer cells primarily depending on the BCL-2/BAX mechanism [31]. Disruption of this signaling pathway can cause intrinsic resistance to drugs [3]. Therefore, substances targeting multiple factors can enhance the effectiveness of cancer treatment. In the present study, the effects of MB and MAB on the activation of a pro-apoptotic factor, namely caspase-3, were also determined (Figure 4). During the apoptotic process, procaspase-3 is converted to the active form caspase-3, which decomposes proteins, resulting in cell death [3]. Importantly, elevated procaspase-3 has been observed in various cancer cases (e.g., acute myeloid leukemia), involving a poor prognosis [32,33]. Through the regulations of procaspse-3 and cleaved caspase-3 expressions (Figure 4), it may be conjectured that MB and MAB stimulated the conversion of procaspase-3 to caspase-3 in both tested cancer cell lines HL-60 and U266. This may cause the promotion of proteolysis in these cancer cells, which subsequently kills them. In the cell cycle pathway, the activation of the CDK1/cyclin B1 complex plays a major role in the transition from G2 to M phase. Thereby, inhibited CDK1/cyclin B1 activity leads to G2 phase arrest [34]. Several anticancer studies were conducted focusing on G2/M phase arrest through this mechanism. For example, genistein arrested G2/M phase in the cell cycle of colon cancer cells [35] and breast cancer [36]. In our study, the expressions of CDK1 and cyclin B1 in HL-60 were significantly impeded by MB and MAB (Figure 4a). The finding indicates that MB and MAB may disrupt the interaction of CDK1/cyclin B1 complex, causing HL-60 cell cycle arrest at G2 phase, followed by mitosis inhibition. In other studies, the potential of MB in inhibiting colon cancer cells (HT-29 and SW620) was shown via MTT, lactate dehydrogenase (LDH), and colony-forming ability assays [15]. Besides, MB suppressed the human monocytic leukemia cell line U937 by stimulating apoptosis and the cell cycle arrest at G1 phase via the decrease in pRB phosphorylation and the upregulation of a CDK inhibitor p21^Waf1/Cip1^ [13]. Lee et al. [14] announced that MB prevented human leukemic T cells (Jurkat) by inducing apoptosis through the mitochondrial pathways. In addition, the inhibitory effect of MB on HL-60 cell viability was previously demonstrated [14], but the cytotoxic mechanism has not been elucidated. Furthermore, to the best of our knowledge, the present study is the first to clarify the induction of MB and MAB on apoptotic and cell cycle arrest pathways of HL-60 and U266 cells through the regulation of relevant proteins (Figure 5). Based on evidence from in vitro assays, we highlighted the cytotoxic potential of momilactones on HL-60 and U266 cell lines by comparing to that of well-known medicines (doxorubicin, ATRA, ATRA/ATO, and bortezomib), which have not been reported elsewhere (Figure 1, Table 1, and Appendix A). However, the pharmacodynamics of drugs and tested agents are variable depending on the mechanism of cytotoxic action and multiple factors such as drug uptake, intracellular metabolism, interaction with target molecules, and efflux from the cell [37]. Therefore, the actual effectiveness of momilactones in preventing APL and MM requires deeper clarification and confirmation through pharmacodynamic as well as pharmacokinetic studies to develop them as novel anticancer drugs. Besides, the comparison between momilactones and well-known medicines should be investigated. For example, in the case of APL therapies, all-trans retinoic acid (ATRA), and the combination of ATRA and arsenic trioxide (ATRA/ATO) are widely used [20]. While for MM treatments, proteasome inhibitors (PI) (e.g., bortezomib, carfilzomib, and ixazomib), immunomodulatory agents (IMiD) (e.g., lenalidomide, pomalidomide, and thalidomide), monoclonal antibodies (e.g., daratumumab and elotuzumab), and targeted B cell maturation agent (BCMA) therapies are commonly applied [19]. In our research, ATRA, ATRA/ATO, and bortezomib were tested (Figure 1, Table 1, and Appendix A), whilst other mentioned medicines were neither available in our laboratory nor purchased, and thus need further investigation. Moreover, the combined use of momilactones and these drugs should be investigated, aiming to enhance the efficiency of targeted therapies, reduce the negative side effects, and overcome drug resistance [19,20,38,39]. On the other hand, the impacts of momilactones on normal cells should be comprehensively interpreted since they might be affected by the same course of events with tumors [40]. In this study, momilactones slightly promoted cell apoptosis of the normal mesothelial cell line (MeT-5A) but exhibited no effects on the cell cycle. Forthcoming studies should focus on the alterations of relevant proteins to the apoptotic process in MeT-5A cells treated with momilactones. Additionally, other cells sensitive to drugs, including bone marrow, gonads (sex organs), gastrointestinal tract, and skin (hair follicle cells), should also be included to clearly understand the toxicity or adverse effects of momilactones [40]. This is an integral requirement to minimize failure in later stages of drug development. Furthermore, this may help establish potential drug combinations as well as effective therapies [40].

In addition to the above-mentioned anticancer potentials, the antioxidant capacity of momilactones was previously reported. It was noteworthy that the synergistic effect of MA and MB revealed a stronger antioxidant capacity than individual compounds [18]. This may address the complications in cancer progression and treatment due to the negative side effects of drugs associated with oxidative stress [21]. Thus, substances simultaneously revealing antioxidant and cytotoxic properties may be excellent candidates for the development of effective cancer therapies. Moreover, a correlation between chronic disorders, including diabetes, obesity, aging, and cancer, through the central role of inflammation and oxidative stress has been acknowledged [41]. Interestingly, momilactones have recently exhibited potential for anti-diabetes, anti-obesity, and anti-skin aging activities [16,17,18]. Thereby, these compounds can be considered a promising source for improving blood cancer treatments, especially for patients complicated with oxidative stress and chronic diseases.

MA and MB were principally found in rice husk [42], leaf [43], and root [42]. Recently, a specific sample preparation technique and advanced ultra-performance liquid chromatography-electrospray ionization-mass spectrometry (UPLC-ESI-MS) method were improved to increase the detection sensitivity that help quantify MA and MB in different rice organs with a minor amount (e.g., in rice bran) [16]. Those outstanding achievements may support prospective strategies to take advantage of momilactones from rice and rice by-products for pharmaceutical purposes. Notably, rice is a monocot plant, which adapts to a wide range of environmental conditions [44]. Thus, rice organs can be feasibly exploited for medicinal production and therapeutics with an abundant biomass availability. In addition to momilactones, 47 momilactone-like molecules have been acknowledged [12], suggesting an abundant source for further investigations of their cytotoxic potentials against blood cancer cells. As another concern, the biological activity and bio-accessibility of substances can be affected by human digestion [7]. Accordingly, future studies should be conducted to investigate the bio-accessibility and bioavailability of momilactones during the digestive stages. Moreover, a natural-based product must satisfy the requirements of benefits outweighing risks [10]. Therefore, the effective concentration of momilactones should be established to exhibit the strongest cytotoxicity against tumors without harmful effects on normal cells. On the other hand, the potential risks, such as neurotoxicity and hepatotoxicity, of using herbal products should be seriously considered and thoroughly evaluated [38].

In summary, MB and MAB are promising candidates, which are highly recommended for the next steps of developing novel anti-APL and anti-MM medicines. In vivo tests are required to confirm their possibilities for proposing prospective and appropriate clinical trials.

## 5. Conclusions

In this report, we interpreted, for the first time, the cytotoxic mechanism of momilactones A (MA) and B (MB) and their mixture (MAB) against acute promyelocytic leukemia (APL) HL-60 and multiple myeloma (MM) U266 cell lines. Based on the evidence from in vitro assays, MB and MAB substantially inhibited the cell viability of HL-60 and U266, with an IC_50_ of around 5 µM. Especially, the cytotoxicity of MB and MAB against HL-60 was in line with that of the well-known medicines doxorubicin, all-trans retinoic acid (ATRA), and the mixture of ATRA and arsenic trioxide (ATRA/ATO). Besides, MB and MAB may induce HL-60 and U266 cell apoptosis via the mitogen-activated protein kinase (p-38) and mitochondrial (BCL-2 and caspase-3) signaling pathways. In addition, HL-60 cell cycle was arrested at G2 phase by MB and MAB through the regulations of related protein (p-38, CDK1, and cyclin B1) expressions. Significantly, momilactones revealed a slight effect on the normal cell line MeT-5A. It can be concluded that momilactones are promising candidates for developing novel anti-APL and anti-MM medicines. Moreover, momilactones and momilactone-like compounds are expected as prospective natural sources for future pharmarceutical production and therapeutics. However, the dose-effectiveness, bio-accessibility, and bioavailability of these analytes need validation via in vivo tests before considering further clinical trials.

## Figures and Tables

**Figure 1 cancers-14-04848-f001:**
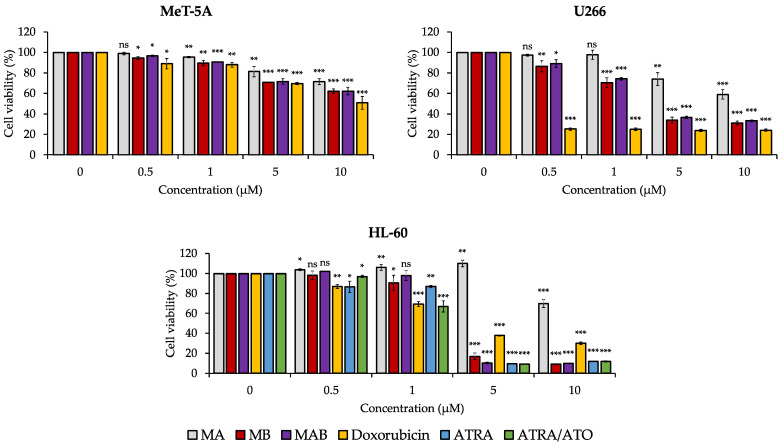
Effects of momilactones A (MA) and B (MB) and their mixture (MAB) (1:1, *w*/*w*) on cell viability of non-cancerous (MeT-5A), acute promyelocytic leukemia (HL-60), and multiple myeloma (U266) cell lines after 48 h. Data are expressed as mean ± standard deviation (SD). Statistical significance was determined by the *t*-test. * *p* < 0.05 versus control (0 µM), ** *p* < 0.01 versus control (0 µM), *** *p* < 0.001 versus control (0 µM). ATRA, all-trans retinoic acid; ATRA/ATO, the mixture of all-trans retinoic acid and arsenic trioxide (1:1, *w*/*w*); ns, not significant versus control (0 µM).

**Figure 2 cancers-14-04848-f002:**
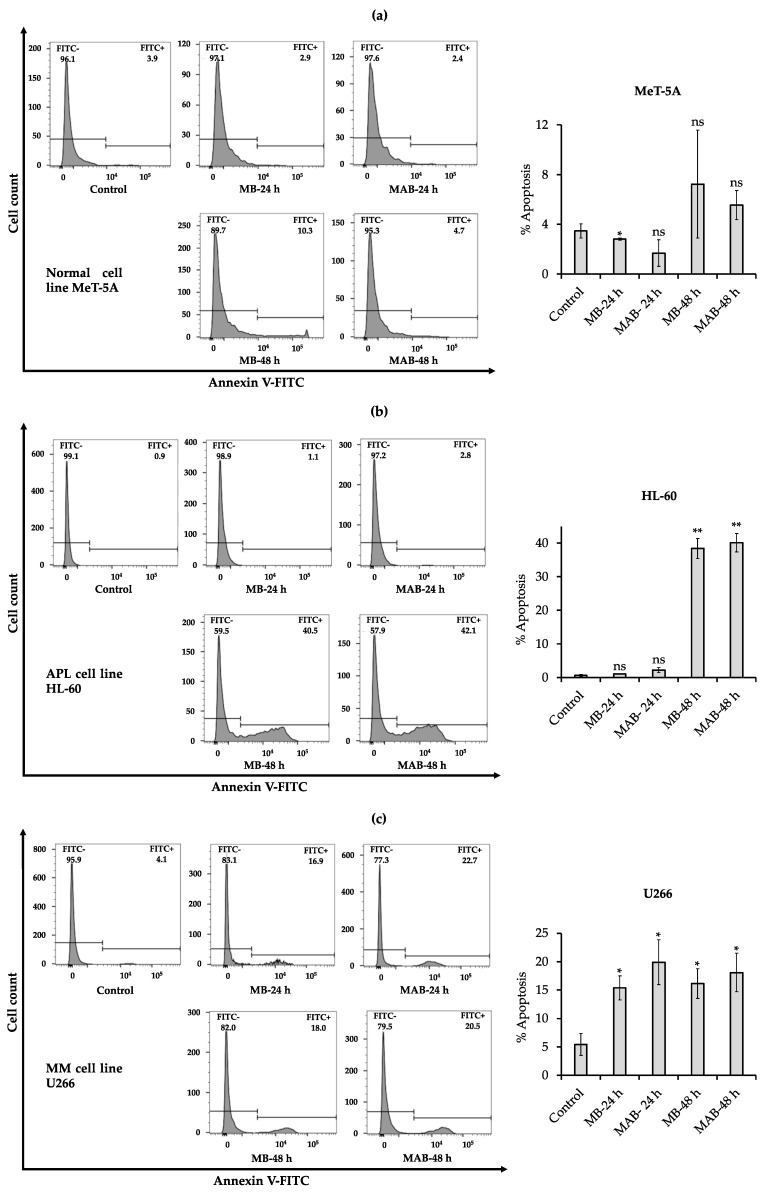
Apoptosis-inducing effects of momilactone B (MB) and the mixture of momilactone A and B (MAB) (1:1, *w*/*w*) at 5 µM against (**a**) non-cancerous MeT-5A, (**b**) acute promyelocytic leukemia (APL) HL-60, and (**c**) multiple myeloma (MM) U266 cell lines after 24 and 48 h. Statistical significance was determined by the *t*-test. * *p* < 0.05 versus control, ** *p* < 0.01 versus control. ns, not significant versus control.

**Figure 3 cancers-14-04848-f003:**
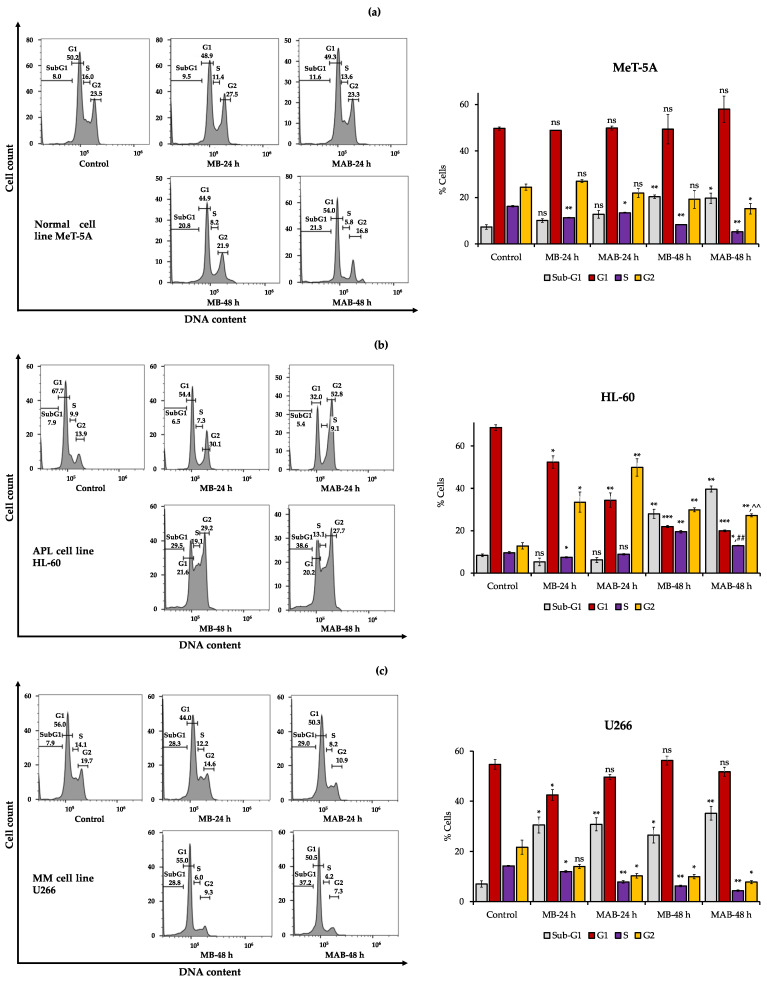
Effects of momilactone B (MB) and the mixture of momilactone A and B (MAB) (1:1, *w*/*w*) at 5 µM on the cell cycle of (**a**) non-cancerous (MeT-5A), (**b**) acute promyelocytic leukemia (HL-60), and (**c**) multiple myeloma (U266) cell lines after 24 and 48 h. Statistical significance was determined by the *t*-test. * *p* < 0.05 versus control, ** *p* < 0.01 versus control, *** *p* < 0.01 versus control, ^^^^
*p* < 0.01 versus MAB-24 h, ^##^
*p* < 0.01 versus MB-48 h. ns, not significant versus control.

**Figure 4 cancers-14-04848-f004:**
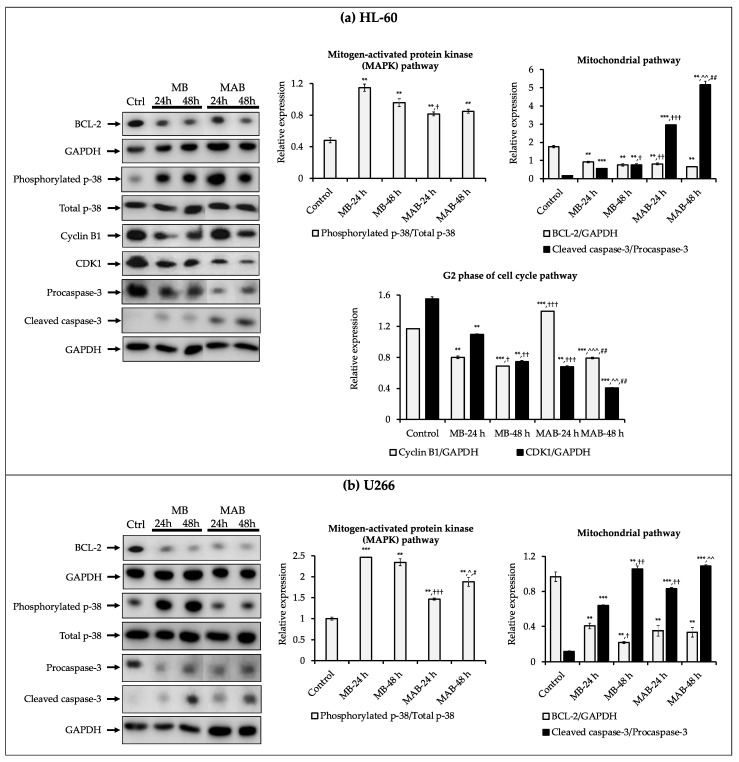
Effects of momilactone B (MB) and the mixture of momilactones A and B (MAB) (1:1, *w*/*w*) on the expressions of proteins related to apoptosis induction and cell cycle arrest of (**a**) acute promyelocytic leukemia (APL) HL-60 and (**b**) multiple myeloma (MM) U266 cell lines after 24 and 48 h. Statistical significance was determined by the *t*-test. ** *p* < 0.01 versus control; *** *p* < 0.001 versus control; ^†^
*p* < 0.05 versus MB-24 h; ^††^
*p* < 0.01 versus MB-24 h; ^†††^
*p* < 0.01 versus MB-24 h; ^^^
*p* < 0.05 versus MAB-24 h; ^^^^
*p* < 0.01 versus MAB-24 h; ^^^^^
*p* < 0.001 versus MAB-24 h; ^#^
*p* < 0.05 versus MB-48 h; ^##^
*p* < 0.01 versus MB-48 h. The uncropped blots are shown in Appendix A.

**Figure 5 cancers-14-04848-f005:**
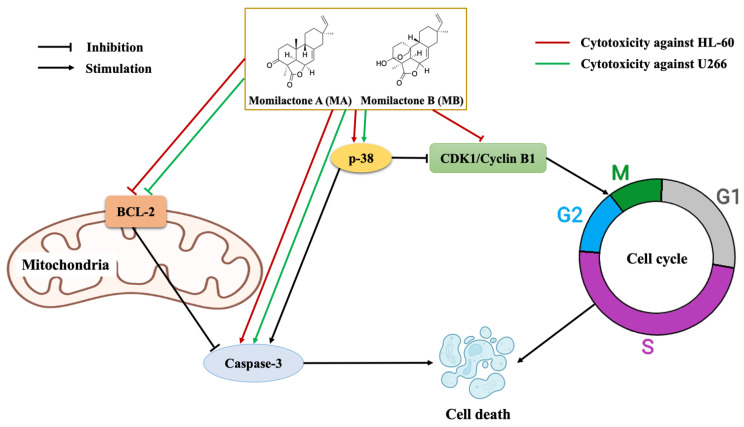
Cytotoxic mechanism of momilactones (**A**) (MA) and (**B**) (MB) against acute promyelocytic leukemia (APL) HL-60 and multiple myeloma (MM) U266 cell lines.

**Table 1 cancers-14-04848-t001:** Cytotoxic activities of momilactones A and B against MeT-5A, HL-60, and U266 cell lines.

Compounds	MeT-5A (% Inhibition at 10 µM)	HL-60 IC_50_ (µM)	U266 IC_50_ (µM)
MA	28.52 ± 2.93 ^c^	-	-
MB	38.00 ± 2.29 ^b^	4.49 ± 0.34 ^bc^	5.09 ± 0.58 ^a^
MAB	37.82 ± 3.64 ^b^	4.61 ± 0.10 ^b^	5.59 ± 0.17 ^a^
Doxorubicin	49.23 ± 6.17 ^a^	5.22 ± 0.15 ^a^	0.24 ± 0.01 ^b^
ATRA	-	3.99 ± 0.16 ^cd^	-
ATRA/ATO	-	3.67 ± 0.20 ^d^	-

Outcome is presented as mean ± standard deviation (SD). Means within a column followed by similar superscript letters (^a,b,c,d^) are insignificantly different at *p* < 0.05 (one-way ANOVA). MA, momilactone A; MB, momilactone B; MAB, the mixture of MA and MB (1:1, *w*/*w*); ATRA, all-trans retinoic acid; ATRA/ATO, the mixture of all-trans retinoic acid and arsenic trioxide (1:1, *w*/*w*); -, not determined.

## Data Availability

Data are contained within this article.

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
