# Peer review of "Cytotoxic Mechanism of Momilactones A and B against Acute Promyelocytic Leukemia and Multiple Myeloma Cell Lines"

_cancers, 2022, doi:10.3390/cancers14194848_

Round 1

Reviewer 1 Report

Anh et al reported on the use of momilactone in APL and MM cell lines (HL-60 and U266). The rationale of this study is interesting. However, the authors did not understand the management of APL and MM in the clinical practice, making the results irrelevant in this field. Doxorubicin is not the primary therapy for APL (Sanz et al. Blood 2019), and ibrutinib is even not approved for the treatment of MM (Rajkumar SV. Am J Hematol. 2022). Therefore, the design of this study is not appropriate. The authors should compare MAB with ATRA/ATO for APL, and with PI(bortezomib, carfilzomib, ixazomib)/IMiD (lenalidomide, pomalidomide, thalidomide)/daratumumab/BCMA-targeted therapies for MM.

Author Response

Dear Respective Reviewer 1

Thank you very much for your valuable comments and suggestions to improve our manuscript. We have revised it carefully, noted by red color letters in the revised manuscript, and noted where and how we revised it in the attached file. Please kindly check.

Thank you very much

Tran Dang Xuan

Reviewer 2 Report

Thank you very much for providing an opportunity to review the article titled “ Cytotoxic Mechanism of Momilactones A and B against Acute  Promyelocytic Leukemia and Multiple Myeloma Cell Lines” by La Hoang Anh and co-authors. 

In this manuscript La Hoang Anh and co-authors tested for the first time the cytotoxic mechanism of rice derived phytoalexins mamilactone A and B in the promyelocytic leukemia and multiple myeloma cell lines HL-60 and U266 respectively. They have used the noncancerous cell line MeT-5A as a control. They have employed cell viability, apoptosis, cell cycle analysis. Additionally, western blot was used to detect apoptosis and cell cycle arrest signaling in these cell lines. The authors have shown that mamilactone A, B or its combination increases apoptosis and induce cell cycle arrest in these cells. 

Major comments:

  1. The authors failed to mention the statistical methodology used to interpret the results.
  2. For the cell viability, apoptosis or the cell cycle assay, the authors fail to mention the concentration of cells used. 
  3. In the figures showing the cytotoxicity of mamilactones (Fig 1), for the HL-60 cell line, ibrutinib was not used as a control. Is there a particular reason for that. Please discuss it in the text. The y-axis of the Fig.1 and the text is not matching, Please keep the y-axis to show the percentage inhibition with respect to 100% . In addition, please mention the statistical method used to analyze the differences observed between drugs. Indicate the p-values in the figure.
  4. For the apoptosis analysis (Fig.2), it would be better to include a bar graph and statistics to clearly understand the differences between treatment and time in these cell lines. Please mention the statistical method used to measure the significance and indicate the p-value in the figure. 
  5. Similarly, for the cell cycle analysis (Fig.3), please indicate the statistical method used to address the differences between the treatment and time in these cell lines. Preferably a bar graph showing the % population of cells in each cycle comparing different treatment and time for each cell lines would be good to make it easy to compare the results.
  6. In the western blot analysis, the authors have used one-way ANOVA as a statistical method to compare the treatments and time. It would be ideal to use a t-test to compare each time points for each drug to control to get a better sense of statistical differences.

Author Response

Dear Respective Reviewer 2

Thank you very much for your valuable comments and suggestions to improve our manuscript. We have revised it carefully, noted by red color letters in the revised manuscript, and noted where and how we revised it in the attached file. Please kindly check.

Thank you very much

Tran Dang Xuan

Reviewer 3 Report

In the paper " Cytotoxic Mechanism of Momilactones A and B against Acute 2 Promyelocytic Leukemia and Multiple Myeloma Cell Lines” La Hoang Anh et al.,  provides new information on the cytotoxic mechanisms of momi-378 lactones A (MA) and B (MB) and their mixture (MAB) against acute promyelocytic leuke-379 mia (APL) HL-60 and multiple myeloma (MM) U266 cell lines. Although  the manuscript is well written and the results are interesting I think that they are weak.  The authors should be explain better the results, for example why is there  an increase of  cells in  subG1 in the “normal” cells? although no effect on cell death was observed?. More,

Authors should represent the apoptosis results more clearly. It is clear that annexin/PI  assay is able to identify 4 cell population: live (annexin and PI negative cells), early apoptosis (annexin pos e PI negative cells), late apoptosis (annexin and PI positive cells) and necrotic cells (annexin negative e PI positive cells), so the cells that they called necrotic are not necrotic but apoptotic. Please reframe the apoptosis part including the figure. About western blot, please add the results obtained in normal cell line. 

Author Response

Dear Respective Reviewer 3

Thank you very much for your valuable comments and suggestions to improve our manuscript. We have revised it carefully, noted by red color letters in the revised manuscript, and noted where and how we revised it in the attached file. Please kindly check.

Thank you very much

Tran Dang Xuan

Reviewer 4 Report

Major points :

-Figure 4a and 4b: I am very surprised that the cleaved form is so present in the control? Moreover, this is not explained.

-Figure 4a and 4b: the authors show only the inactive form of p38. Indeed, it is a major lack of scientific rigour not to show the expression of the phosphorylated form representing the active form of MAPKs. For example, in a recent work on the same cells (HL-60, see figure 11 of the publication by Rodrigues et al. on the activities of triterpenes: Biomed Pharmacother. 2021 Oct;142:112034. doi: 10.1016/j.biopha.2021.112034), the phosphorylated forms of the MAPKs (p38, JNK and ERK) are analysed. These experiments can be analysed either by Western blot or by ELISA.

-Finally, this work presents few results for a journal with an impact factor of 6.575 and moreover unoriginal results.

Minor point :

-Both in the legend of figure 1 and in the results the processing time is missing.

Author Response

Dear Respective Reviewer 4

Thank you very much for your valuable comments and suggestions to improve our manuscript. We have revised it carefully, noted by red color letters in the revised manuscript, and noted where and how we revised it in the attached file. Please kindly check.

Thank you very much

Tran Dang Xuan

Round 2

Reviewer 1 Report

I thank the authors for the revised version. However, I don't think the selection of "reference inhibitors” was appropriate. As the authors have shown in Figure 1/S1, ibrutinib has pretty low cytotoxic effect in MM cell line U266, and MAB seem to have a lower IC50 than bortezomib. But why? Ibrutinib, bortezomib, and doxorubicin have completely different pharmacodynamics. Moreover, this manuscript contained many strong conclusions (e.g. the statement regarding further tests of MAB within clinical trials), which were not supported by the few results based on experiments with a single cell line of each entity. Furthermore, the comparison MAB vs ATRA in APL is still missing in the revised version. For a journal with a “middle-class” impact factor (6.57), it is difficult for me to endorse the publication here. I will leave the decision in charge of the editors.

Author Response

Dear Respective Reviewer 1

Thank you for your constructive comments and suggestions. We have revised the paper and answer  in details of each queries in the attached file. We also noted by red color letters in the manuscript and noted where and how we revised and improved our paper. We do hope that our revision will be able to satisfy your comments and requests.

Thank you again 

Tran Dang Xuan

Reviewer 2 Report

The authors have adequately addressed the concerns, thus improved the quality of the presentation. I would recommend the manuscript to be accepted in the current form.

Thank you very much for providing an opportunity to review this work.

Author Response

Dear Respective Reviewer 2

Thank you very much for your kind constructive comments and suggestions.

Best regards

Tran Dang Xuan

Reviewer 3 Report

Dear Authors,

thank you for addressing all my comments.  

Author Response

Dear Respective Reviewer 3

Thank you for your constructive comments and suggestions to improve our paper. We have some revisions and noted by red color letters in the manuscript. We do hope that our revision will be able to further satisfy your comments and requests.

Thank you again 

Tran Dang Xuan

Reviewer 4 Report

I had proposed a rejection of the publication. In view of the new results presented, as much as I am convinced of the activation of p38, I find the new cleaved caspase-3 western blots of poor quality. I request that the original western blots be sent with each time a molecular weight marker that migrated at the same time as the tested samples.

Author Response

Dear Respective Reviewer 4

Thank you for your constructive comments and suggestions. We have revised the paper and answer  in details of each queries in the attached file. We also noted by red color letters in the manuscript and noted where and how we revised and improved our paper. We do hope that our revision will be able to satisfy your comments and requests.

Thank you again 

Tran Dang Xuan

Round 3

Reviewer 1 Report

I thank the authors for the revised manuscript. Currently, APL patients receive the combination ATRA/ATO, as ATO monotherapy is less effective and is not the standard anymore. The authors demonstrated that MAB has similar anti-APL activity as ATRA, which induces the cell maturation of leukemic blasts. Therefore, the comparason MAB vs ATO mono is not necessary, and MAB vs ATRA/ATO combination should be enough. Obviously, MAB did not show any significant advantage when compared to the combination ATRA/ATO. In addition, the authors stated that inhibition of BCL-2 might be a potential mechanism of MAB. Do the U266 cells present t(11;14)? Moreover, I noticed that the authors did not remove the comparison MAB vs ibrutinib in MM. Why? Ibrutinib is not an anti-MM drug, and the design of this experiment is not apropriate. How about the "normal" anti-MM drugs lenalidomide, pomalidomide, carfilzomib, and daratumumab etc?  In summary, this study appears really strange especially for the clinicians. I would recommend refusal of this manuscript and the authors could try to submit it to a non-medical journal. As mentioned in my previous review report, I will leave the decision in charge of the editors.

Author Response

Dear Respective Reviewer

Thank you for your valuable comments to increase the quality of our manuscript. We have revised the paper carefully, adding with supplemented experiments with data, please check in the attached file. We do hope this time, it can satisfy you.

Thank you again

Best regards

Tran Dang Xuan

Reviewer 4 Report

Now, this work is acceptable for publication in Cancers.

Author Response

Dear Respective Reviewer 4

Thank you for your valuable comments to increase the quality of our paper. We have revised it carefully. Please check in details how we answered and where we revised our manuscript in the attached file.

Thank you very much

Tran Dang Xuan

Round 4

Reviewer 1 Report

I thank the authors for the revision! This manuscript has a sufficient quality for publication in cancers.